# Slow Release and Water Retention Performance of Poly(acrylic acid-*co*-acrylamide)/Fulvic Acid/Oil Shale Semicoke Superabsorbent Composites

**DOI:** 10.3390/polym14091719

**Published:** 2022-04-22

**Authors:** Yongsheng Wang, Yongfeng Zhu, Yan Liu, Bin Mu, Aiqin Wang

**Affiliations:** 1Key Laboratory of Clay Mineral Applied Research of Gansu Province, Center of Eco-Material and Green Chemistry, Lanzhou Institute of Chemical Physics, Chinese Academy of Sciences, Lanzhou 730000, China; wysh0304@126.com (Y.W.); liuyanlwb@163.com (Y.L.); mubin@licp.cas.cn (B.M.); 2Center of Materials Science and Optoelectronics Engineering, University of Chinese Academy of Sciences, Beijing 100049, China

**Keywords:** superabsorbent composite, fulvic acid, semicoke, slow release, plant growth

## Abstract

In order to achieve the low cost and multifunction of superabsorbent composites, poly(acrylic acid-*co*-acrylamide)/fulvic acid/oil shale semicoke (PAMFS) were prepared by free radical copolymerization of fulvic acid (FA), oil shale semicoke (OSSC), acrylic acid (AA) and acrylamide (AM). The characterization results revealed that FA and OSSC were involved in the construction of a three-dimensional (3D) polymeric network via hydrogen bonding and covalent bonding. The water absorbency of PAMFS in distilled water and 0.9 wt% NaCl solution were 724 and 98 *g*/*g*, respectively. The FA slow release of PAMFS in distilled water and soil was achieved due to the interaction between FA and the functional groups of polymer matrix by hydrogen bonds and covalent bonds. Furthermore, the potted experiment indicated that the addition of PAMFS to soil can significantly promote plant growth compared with the pure soil, regardless of water stress. Therefore, this superabsorbent composite showed an excellent water absorption and salt resistance performance, as well as nice slow release performance. It has a broad application prospect.

## 1. Introduction

The climatic changes and the consequent land desertification and global water shortage have sounded the alarm for agricultural production and food crisis [1]. Many innovative technologies or materials are also being developed to solve these global issues [2,3,4]. Among them, the superabsorbent, which own the physically and chemically crosslinked polymeric structure, have attracted attention as the material capable of absorbing and releasing water [5,6,7]. Recently, the novel superabsorbent containing the slow release performance set off a new research upsurge. Except for the role of conservation water, lowering surface runoff and avoiding soil erosion, the unique feature of slow release performance could promote crop growth or suppress weed growth [8,9]. Even so, the current approach for preparation of slow release superabsorbent still needs multiple steps, and the complex process not only significantly increases the cost, but also the low salt tolerance cannot adapt to the complex soil environment. Therefore, realizing the slow release of fertilizer and high salt tolerance by reaction via a one-step process has important implications [10,11,12].

Humic acid (HA), which is known as the “sleeping giant” hidden in the ground, is a natural polymer with high functionality [13]. Fulvic acid (FA) is a kind of water-soluble polymer with the smallest molecular weight, the highest physiological activity and a variety of functional groups in HA. It can promote nitrogen uptake by plants and prevents soil fixation of phosphorus and potassium, promote the formation of soil aggregates and improve the physical and chemical properties of soil. FA can also enhance photosynthesis, promote crop growth, regulate stomatal openness, reduce transpiration and improve drought resistance [14]. HA has been used as a component of network structures in prepared superabsorbent materials in the past decade. For instance, Araujoet al. [15] prepared a novel multifunctional superabsorbent by using chitosan and HA via electrostatic forces between the NH_3_^+^ and COO^−^ groups to control the release of urea dispersed in chitosan. Shenet al. [16] reported a fertilizer with nitrogen slow release properties by combining urea and HA modified bentonite; the results showed the fertilizer in the soil could increase yield and nitrogen uptake of wheat. However, less attention has been paid to the application of FA in superabsorbent materials, although it has greater advantages than HA to improve the comprehensive properties of composite superabsorbent. Therefore, the preparation of a new type of slow release fertilizer based on the physico-chemical properties of FA is not only of great significance to the efficient utilization of FA, but also can significantly reduce agricultural costs.

In our previous work [17], a novel PAA/FA/OSSC was prepared by copolymerizing AA, FA and OSSC, which demonstrated OSSC and FA can effectively improve the water absorption of superabsorbent composite and realize the slow release of FA in water and soil. However, there is still room for further improvement in the slow release performance. In this work, a new type of superabsorbent composite of poly(acrylic acid-*co*-acrylamide)/fulvic acid/oil shale semicoke (PAMFS) was prepared. We expect the introduction of non-ionicpolymerization monomer AM can improve the salt resistance and slow release performance of superabsorbent composites. The polymeric mechanism was analyzed systemically and the water absorption and retention performance as well as the slow release effect of FA were studied carefully.

## 2. Experimental

### 2.1. Materials

Oil shale semicoke (OSSC, milled and filtrated through 200-mesh screen) was received from Yaojie Coal and Electricity Group Co., Ltd. (Gansu, China). Fulvic acid (FA, chemically pure, ≥85%, ash = 10%, moisture < 8%, Fe ≤ 0.3%) was obtained from Tianjin Guangfu Fine Chemical Research Institute (Tianjin, China). Acrylic acid (AA, chemically pure) and acrylamide (AM, chemically pure) were purchased from Tianjin Kermel Chemical Factory (Tianjin, China). *N*,*N’*-methylene bisacrylamide (MBA, chemically pure) and ammonium persulfate (APS, analytical pure) were purchased from Chengdu Chron Chemical Factory (Sichuan, China). Sodium hydroxide (NaOH, chemically pure) was purchased from Rionlon Bohua (Tianjin, China) Pharmaceutical & Chemical Co., Ltd. (Tianjin, China). Other chemical reagents were analytical grade and utilized as obtained without further purification.

### 2.2. Synthesis of Superabsorbent Composites

The superabsorbent composite with the FA slow release performance was synthesized by a convenient free radical reaction. Typically, 0.98 g OSSC and 1.28 g FA were placed into the three-necked flask containing 30 mL distilled water, the mixture solution was stirred vigorously and dispersed uniformly at 75 °C in an oil bath while being flushed with nitrogen for 30 min to remove oxygen from the system. Subsequently, 5 mL aqueous solution containing 0.14 g APS was charged into the above flask and kept for 2 min. Next, 0.03 g MBA, 3.27 g AM and 3.93 g AA were mixed well with a magnet and slowly added to the above reaction system. Finally, the polymerization reaction was completed, the stirring was stopped and maintained for 2 h at 75 °C. The product was dried at 90 °C to constant weight, crushed and classified into particle sizes of 40–60 mesh for further experiment. The superabsorbent composites of PAMS without FA were obtained according to the above preparation procedure.

### 2.3. Characterization

The polymeric structures of PAMFS and PAMS were characterized by using Thermo Nicolet NEXUS FTIR spectrometer (Thermo Fisher Scientific, Wilmington, DE, USA). The surface morphologies of PAMFS and PAMS were observed by the field emission scanning electron microscope (FE-SEM, JSM-6701F, JEOL, Tokyo, Japan) after coating the samples with gold film. The thermal stability of PAMFS and PAMS was evaluated with the STA449C thermo gravimetric analyzer (Perkin-Elmer Co., Groningen, The Netherlands), and the samples were heated from 30 to 700 °C with a rate of 10 °C/min under nitrogen atmosphere.

### 2.4. Swelling Kinetics of Superabsorbent Composites in Water

The water absorbencies of PAMFS and PAMS in different solution media were calculated via the gravimetric method. First, 0.05 g dried superabsorbent composites (W_1_) were immersed in 400 mL solution media and soaked for certain time intervals (300 s, 600 s, 900 s, 1800 s, 2700 s and 3600 s)at room temperature. Subsequently, the swelling samples were placed on a sieve (100 mesh) and the redundant aqueous solution was removed then weighed accurately (W_2_). The equilibrium swelling capacity (Q_eq_, *g*/*g*) of the superabsorbent composites was calculated using Equation (1) [18]:Q_eq_ = (W_2_− W_1_)/W_1_(1)

### 2.5. Reswelling Capability and Retention Behavior in Distilled Water and Soil

The re-swelling capability of PAMFS and PAMS in distilled water was examined through six repetitions of the “swelling–drying” cycle [19]. Typically, 0.1 g sample was immersed in 400 mL distilled water for 4 h to reach the swelling equilibrium, then the swelling sample could be completely dehydrated in an oven at 90 °C. The dehydrated sample was reimmersed to determine its equilibrium swelling capacity. The Q_eq_ of every time swelling was calculated using Equation (1). 

The water-retention capacity of PAMFS and PAMS in soil was investigated. Typically, 100 g dried sandy soil was blended with superabsorbent and transferred into a plastic bottle that was very porous at the bottom, and the total weight was recorded as W_0_. Then, the plastic cup was placed in a tray filled with water and left for a certain amount of time to ensure water saturation. The plastic bottle was weighed and marked as W_2_. Recording the bottle weighed at the standing time t and counted as Wt, the water-retention ratio was obtained according to Equation (2) [20].
Water retention= ((W_t_ − W_0_)/(W_2_ − W_0_)) × 100%(2)

### 2.6. Slow Release Behavior in Water and Different Salt Concentrations

The dried sample (0.5 g) was first placed in 400 mL of distilled water or saline solutions (KCl, MgCl_2_ and AlCl_3_), stirring constantly. Then, 5 mL of solution was taken from the beaker at determined time intervals. Meanwhile, 5 mL of distilled water or the saline solutions were added, with the aim of constantly maintaining the volume of the culture solution. The release of carbon content was determined by the chemical oxygen demand method (COD). The mechanism is that the FA and other organic matter are all oxidized with K_2_Cr_2_O_7_ under concentrated sulfuric acid conditions, and the amount of FA and other organic matter is determined by testing the concentration of produced Cr^3+^ via UV spectra [21,22]. The released amount of the FA was obtained according to Equation (3) [19].
C_FA_ (mg/L) = C_t_ − C_0_(3)
where the C_FA_ is the COD of the FA, C_t_ stands for total organic matter and C_0_ is the COD of others organic matter at different slow release times.

### 2.7. Soilcolumn Leaching Experiment

Typically, the bottom of the organic tube would be filled with 40 g of fine quartz sand. Later, the 400 g sandy soil sample (100 mesh) was thoroughly mixed with superabsorbent and placed in the upper layer of fine quartz sand, respectively. Subsequently, the upper layer of each soil column was covered with quartz sand to reduce the water flow impact pressure on the leaching process. After filling the soil pillars, 250 mL of tap water was irrigated to fully wet the soil column until it was impermeable with tap water and the soil columns were incubated for 24 h. After this, 250 mL of tap water was irrigated after 1, 5, 10, 15, 20, 25, 30, 35, 40 and 45 days, and the leaching solution was collected. The content of organic carbon in the leaching solution was calculated by the COD method.

### 2.8. Pot Culture Experiment

The planting effect of novel superabsorbent was investigated using the Qingan cabbage [23]. Typically, 4 g PAMFS or PAMS was mixed well with 400 g completely dried sandy soil and added into a plastic pot. Subsequently, 20 grains of Qingan cabbage seeds were sown in soil containing superabsorbent composites and germinated under room temperature. The cabbage growing period was maintained for 40 days, during which irrigation was carried out every four or eight days. After harvesting, the roots and seedlings were carefully washed with distilled water and dried to evaluate their growth according to biological characteristics. The control group, the one without adding superabsorbent composite, was carried out according to the above operation sequence.

## 3. Results and Discussion

### 3.1. FTIR Analysis

The polymeric structure of the novel superabsorbent was investigated using FTIR analysis (Figure 1). As showed in Figure 1 line a, the characteristic peaks at 1038, 787, 684, 529, 2924 and 2848 cm^−1^ all derived from the organic carbon and kaolinite of in OSSC [24,25]. Especially, the FTIR stretching peak at 1605 cm^−1^ was due to the bending vibration of hydroxyl or phenolic groups in the organic carbon of OSSC. As can be seen in Figure 1 line b, the peaks in FA spectrum at 1706, 1601 and 1252 cm^−1^ were due to the −C=O stretching vibration of carboxyl groups, asymmetric stretching vibration of carboxylate groups and the stretching vibration of C−OH, respectively [26,27]. After the polymerization, the absorption peaks at 1670 cm^−1^ (the stretching vibration of the CONH_2_), 1564 (the asymmetric stretching vibrations of −COO^−^), 1450 and 1403 cm^−1^ (the symmetric stretching of the −COO^−^) shifted to 1664, 1570, 1443 and 1397 cm^−1^, respectively, as the FA was incorporated with the PAMS (Figure 1 line c). Compared to the FTIR spectra of FA, PAMFS (Figure 1 line d) and the physical mixture of PAMS and FA (Figure 1 line e), the characteristic peaks associated with FA all appeared in the FTIR spectrum of polymers or physical mixtures, but the absorption peak at 1252 cm^−1^ disappeared and the characteristic peaks at 1601 and 1380 cm^−1^ shifted in the FTIR spectrum of PAMFS. By this phenomenon it can be interpreted that the strong chemical bond interactions between functional groups of polymer matrix P(AA-*co*-AM), OSSC and FA may participate in the polymeric network by covalent bond or non covalent bond.

### 3.2. Thermogravimetric Analysis

The chemical structure of PAMFS was further analyzed with the thermogravimetric analysis. As shown in Figure 2, both PAMFS and PAMS revealed four stages of weightlessness at 150–700 °C. The minor mass loss in the first stages at 150–275 °C was attributed to volatilization of organic matter of OSSC (or FA). The second decomposition stages at 275–325 °C corresponded to the weightlessness of bond water in the network and the dehydration of two adjacent carboxyl groups on the polymer chain. The following weightlessness at 325–385 °C was attributed to the breaking of polymeric chains and crosslinked network structure [28].The fourth stage at 385–550 °C gave rise to the decomposition of residual organic matter, and the remaining char fraction amount of PAMFS and PAMS was 52.1% and 45.8%, respectively. It is clear that the maximum weightlessness temperatures of PAMFS in four stages were all higher than PAMS, demonstrating that incorporation of FA into hydrogel matrix can significantly improve thermal stability, and also verifying the covalent bond or non covalent bond interaction between FA and the polymeric network of PAMS [29].

Based on the above characterization result, we can conclude that the 3D network structure of the superabsorbent composite PAMFS was formed according to the following process (Figure 1): the initiator of APS was decomposed by heating and generated sulfur oxide anion radicals, and then the active groups, including the hydroxyl and aldehyde group of in FA and OSSC, were converted into the new radicals by electron transfer reaction [30]. These free radicals acted as active centers and triggered the chain growth reaction in the presence of monomers AA, AM and crosslinker MBA. As the polymeric network formed completely, the FA and OSSC particles were all incorporated into the polymeric network and regarded as the additional cross-linking point [31,32].

### 3.3. SEM Characterization

The surface morphologies of PAMFS are displayed in Figure 3. The flaky particles in PAMS and PAMFS were derived from OSSC, which is due to the fact that its main component was kaolinite. Moreover, the surface of PAMFS showed a looser surface than PAMS. The reason was that the chain entanglement in the polymer matrix became relatively relaxed due to the hindering effect of FA macromolecular structure, which contributed to the rapid infiltration and diffusion of water into the polymer matrix and lead to an increase in water absorption. This result also proved that FA was involved in the construction of the 3D polymeric network. 

### 3.4. Swelling Kinetics of PAMFS

The systematic studies found that the superabsorbent introduced 15 wt% of FA had the highest water absorbencies of 724 and 98 *g*/*g* in distilled water and 0.9 wt% NaCl solution, respectively (Appendix A). Subsequently, the swelling kinetics measurements of PAMS and PAMFS with 15 wt% FA in distilled water and 0.9 wt% NaCl solution were performed and displayed in Figure 4a,b. The swelling capacity of both samples rose dramatically at initial time intervals after immersion into water or 0.9 wt% NaCl solution, and then increased slowly until Q_eq_ was achieved. The PAMFS reached the equilibrium swelling capacity more quickly than PAMS (Figure 4b). Moreover, the Q_eq_ of PAMS was lower than PAMFS. This reason was the hydrophilic hydroxyl groups of FA, which improved the water uptake property of PAMFS and thus enhanced the Q_eq_. Compared with the references on slow-release fertilizers, including NPK and HA, in recent years (Appendix A), the results indicate PAMFS had better water absorption and salt tolerance, as well as more outstanding slow release performance.

To scrutinize the influence of FA on the swelling kinetic behavior, the Scott’s second-order (Equation (4)) was adopted and used to analyze the swelling kinetics of the PAMFS and PAMS [33].
t/Q_t_ = t/Q_∞_ + 1/K_s_Q_∞_^2^(4)
where Q_t_ and Q_∞_ represent the water absorbency at time t(s) and the theoretical equilibrium water absorbency, respectively. K_s_, which corresponds to the structure of the superabsorbent, is the constant rate of swelling. The value (Appendix A) of the diffusion coefficient, including K_s_ and Q_∞_, for the PAMFS and PAMS can be obtained from the slope of these fitting lines. According to the results, the Q_∞_ for PAMFS and PAMS was 729.93 and 632.91 *g*/*g* in distilled water and 105.93 and 75.36 *g*/*g* in 0.9 wt% NaCl solution, respectively, which was very close to the corresponding experimental values. In addition, the K_s_ of PAMFS was lower than PAMS, which demonstrated that the introduction of FA with the mass of functional groups into polymer matrix could form hydrogen bonds with PAMS and improve the crosslinking degree and that it hindered water molecules from penetrating into the polymer network and decreased K_s_ of superabsorbent composites [34].

To evaluate the effect of FA on the water diffusion behavior of PAMFS and PAMS, Equation (5) [35] was utilized for the initial 60% of the water uptake.
F = Q_t_/Q_∞_ = *k*c^n^(5)

This corresponds with the diffusion exponent, which is calculated from the slope of the plots and used to determine the diffusion type of PAMFS and PAMS. The fitting curves of LnF vs. Lnt were shown in Figure 4e,f. For the Fickian diffusion mechanism, the value of *n* must be lower than 0.5, and for non-Fickian diffusion, the value of *n* was between 0.5 and 1.0 [36,37]. The results in Appendix A showed that the values of *n* for PAMFS and PAMS were 0.4679 and 0.5214 in distilled water, which indicated that the introduction of FA in PAMS changed the diffusion type of the water molecule.

### 3.5. Effects of pH on Water Absorbency and Reswelling Capability of PAMFS

The effects of pH on swelling capacity of PAMFS and PAMS were investigated in the pH range of 2–13 at room temperature (Figure 5a). The water absorbency increased with increasing pH from 2 to 6, remained constant in the range of 6–10, and then decreased sharply with further increasing pH. This phenomenon could be explained by the buffer effect of the carboxylate group of the superabsorbent in an acidic or basic solution. In the acidic environment (pH < 6), the protonated carboxylate groups weakened the electrostatic repulsion between the polymer chains and, consequently, decreased the water absorbency. Meanwhile, in a highly basic environment (pH > 10), the increased ionic strength of the solution resulted in the decrease in ion osmotic pressure and reduced swelling capacity.

The swelling capacities of PAMFS and PAMS in distilled water decreased from 724 to 419 *g*/*g* and 619to 320 *g*/*g* after six cycles, losing 42.1% and 48.3% of swelling capacity, respectively (Figure 5b). It was found that PAMFS exhibited lower loss during consecutive swelling/de-swelling cycles compared with PAMS, which, due to the introduction of FA, makes the network structure more stable and enables it to bear the high pressure of absorbed water. Interestingly, our previous studies proved that superabsorbent PAA/FA/OSSC loses 44.1% of swelling capacity after re-swelling six times in distilled water [17], which was higher than PAMFS. They also indicated that the introduction of AM helped to stabilize the network structure.

### 3.6. Slow Release of FA in Distilled Water and Salt Solutions

From the reaction mechanism (Figure 1), FA and OSSC were introduced into the P(AA-*co*-AM) network structure, which cannot only physically fill in the polymeric network through hydrogen bonding, but also can be chemically bonded with the polymer chain through the active groups and participate in the construction of 3D network. Recent studies pointed out that the introduced FA or organic matter derived from OSSC can be released slowly into water or soil as organic fertilizer to promote plant growth [12,17]. The release amounts of PAMFS, adding 5, 15 and 25 wt% of FA, were rapid in the initial stage and then gradually became flat after 20 days in distilled water; the carbon content reached to 23.71, 49.12 and 63.20 mg/L, respectively (Figure 6a). This result indicated that the released amount of FA was not completely determined by the water absorbency of PAMFS, which also confirmed our conjecture about the polymerization mechanism. In the early stage, the FA interacted with the polymeric network via the weaker noncovalent bonding, such as the hydrogen bond, and hydrophobic effect may be released fastest. Subsequently, the release rate slowed significantly, mainly due to the stronger interaction between the FA and polymeric network, for instance, the electrostatic interaction and covalently bonding, etc. [38].

Furthermore, the introduction of OSSC particles also had a positive effect in the release of FA due to the role of auxiliary crosslinking and the more tortuous paths in the superabsorbent, which not only extended the 3D network structure, but also improved the water absorption performance of the superabsorbent [39]. The effects of external solutions such as KCl, MgCl_2_ and AlCl_3_on the slow release of PAMFS (adding15 wt% of FA) were investigated (Figure 6b−d). The released amounts of PAMFS in different saline solutions and salt concentrations were lower than those in distilled water. This phenomenon is mainly due to the tight shrinkage of the polymer network caused by the charge shielding effect of the cations in the salt solution, resulting in the inability of the FA to come into contact with more water and limiting the release of FA. Similarly, the higher salt concentration or ionic charge resulted in the lower osmotic pressure, which prevented water molecules from penetrating inside the polymers and caused a decreased in release efficiency. In brief, the premise of FA’s slow release was contact with water; according to the dynamic exchange of water and release medium and reswelling of the polymer network, FA was gradually released into water via the polymer matrix.

In our previous research work [17], the FA’s slow release of superabsorbent PAA/FA/OSSC in distilled water and salt solution indicated that the release amount of FA was relatively fast in the initial stage and then became flat after 16 days (Appendix A). However, the release platform time of FA release was longer than 16 days with the introduction of AM into PAA/FA/OSSC, especially in salt solution. The reason is that the polymeric network involved in the AM weakens the charge shielding effect of salt ions, causing more water molecules in salt solution to enter the polymer and dissolve more FA.

### 3.7. Water Retention Capacity and Slow Release of Superabsorbent Composites in Soil

Efficient use of rain and irrigation water for plants is significant in arid and semi-arid region. The water-retention capacities of soil with added PAMS and PAMFS (15 wt%) indicated that the water holding capacity of the soil containing 1 wt% of PAMS and PAMFS (15 wt%) was remarkably improved compared with the control group (soil without any superabsorbent) (Figure 7a,c). The moisture of the control group completely evaporated in only 14 days, while the soil samples with added PAMS and PAMFS still retained 27.1 and 38.4% in the same time, respectively. The PAMFS displayed the best water-retention capacity and allowed the water to release out gradually. This may be due to the strong H-bonding interaction between water molecules and FA or organic matter of the OSSC, which retarded the evaporation rate of absorbed water.

The soil column leaching experiment was carried out to simulate the FA’s slow release in soil (Figure 7b,d). The release amount of PAMFS appeared as a trend of first slow and then fast and reached 305.33 mg/L at 20 days. Later, the release amount gradually decreased over time to about 202.13 mg/L after 40 days. The slow release of FA decreased gradually in the later period because the complicated soil environment was more likely to produce “shielding effects” due to the presence of a large number of metal cations. Compared with the FA slow release of PAA/FA/OSSC in soil (Appendix A), there was no rapid decline in the slow release process of PAMFS. This indicated that the more regular network structure and excellent salt resistance performance can improve the nutrient slow release of the superabsorbent composite. The addition of more OSSC may also reduce the collapse probability of the 3D network in a complex soil environment [40].

### 3.8. Pot Culture Experiment

The growth promotion effect of PAMFS for plants was evaluated by using the Qingan cabbage as a target crop [41]. The seeds were embedded into soil and irrigated with an equal quantity of water every 4 days (sufficient irrigation group) or every 8 days (insufficient irrigation group). After 40 days of growth, the seedling showed better growth vigor by addition of PAMFS than PAMS of the blank control group (Figure 8a,b). Subsequently, the fresh plants were harvested and dried at 90 °C (Figure 8c,d), then the biomass of the seedling was determined, and the results were statistically charted in Figure 8e,f. It is obvious that the application of PAMFS was a benefit to plant growth. Under the condition of adequate water, the plant height, root length, fresh weight and dry weight of cabbage seedlings fertilized with PAMFS in soil increased by 28.01%, 24.81%, 143.95% and 182.77%, compared with the blank control group, respectively. Meanwhile, under the condition of limited water, the growth promoting ability of PAMFS was also stronger than that of the blank control group, and the average indexes of the plant height, root length, fresh weight and dry weight increased by 202.77%, 24.35%, 205.90% and 180.07%, respectively. The significant increase in fresh weight directly revealed that the improved crop production can be attributed to the excellent retention of water and slow release performance of FA for PAMFS.

## 4. Conclusions

A novel superabsorbent composite of the PAMFS with outstanding water absorption, salt resistance and slow-release fertilizer performance was synthesized via a simple and green method. The polymerization was confirmed by FTIR, TGA and SEM, which indicated that FA and OSSC participated in the polymer construction by means of chemical bonding and hydrogen bonding. The PAMFS displayed notable water absorbency, being able to absorb distilled water at 724 *g*/*g* and 0.9 wt% NaCl solution 98 *g*/*g*. The diversified polymeric network by copolymerization of AA and AM and involving FA and OSSC led to a complex process of dynamic exchange and swelling diffusion in slow release. The soil column leaching experiment demonstrated that the released amount of the FA and organic matter of OSSC reached 305.33 mg/L after 20 days. In addition, it was found that the introduction of AM could obtain a more stable 3D network structure and improve its salt resistance and slow release performance by comparing PAA/FA/OSSC and PAMFS. The potted results reported that better water management and slow release fertilizer performance of PAMFS could improve the biomass of plants, resulting in the potential application in agriculture.

At present, a large number of multi-functional composite superabsorbents have been reported to solve the problem of plant growth in arid areas. However, the complex preparation process, low salt tolerance and slow release properties have led to high preparation costs. Therefore, on the basis of high value utilization of waste, researchers further reduced production costs and improved comprehensive performance by improving synthesis methods to meet the needs of agricultural production. In this work, PAMFS will provide data support for the application of superabsorbents in agriculture. 

## Data Availability

The data presented in this study are available on request from the corresponding author.

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
