# Peer review of "Slow Release and Water Retention Performance of Poly(acrylic acid-co-acrylamide)/Fulvic Acid/Oil Shale Semicoke Superabsorbent Composites"

_polymers, 2022, doi:10.3390/polym14091719_

Round 1

Reviewer 1 Report

The manuscript entitled, ‘Slow Release and Water Retention Performance of Poly(Acrylic Acid-Co-Acrylamide)/Fulvic Acid/Oil Shale Semicoke Superabsorbent Composites’ reported synthesis superadsorbent hydrogel and its slow release behavior. Though the study is interesting indeed but still I am mentioning some loopholes of this work which should be accounted;

  1. Why fulvic acid is used is not clear. Author should mention this with more emphasis.
  2. Is this hydrogel an interpenetrating network type?
  3. For thermal stability study author should mention the remaining char fraction amount in the text.
  4. Did the author check the material’s delayed water release or deswelling in different temperature?
  5. What was the porosity of the hydrogels?
  6. How humic acid is differed than fulvic acid regarding this delayed water release? Could the author comment on that? No need to do further experiment.
  7. The paper is based on superadsorbent, but there is no as such literature study on hydrogel materials. Therefore, it is recommended to insert some of relevant works like; https://doi.org/10.1016/j.ijbiomac.2016.11.055; https://doi.org/10.1016/j.ijbiomac.2012.07.028; https://doi.org/10.1080/03602559.2016.1233268.        

Author Response

Question 1: Why fulvic acid is used is not clear. Author should mention this with more emphasis?

Author’s Reply: Thanks for your comment! Fulvic acid (FA) is a kind of water-soluble polymers with the smallest molecular weight, the highest physiological activity and a variety of functional groups in humic acid (HA). It can promote nitrogen uptake by plants and prevents soil fixation of phosphorus and potassium, promote the formation of soil aggregates and improve the physical and chemical properties of soil. FA can also enhance photosynthesis, promote crop growth, regulate stomatal open-ness, reduce transpiration and improve drought resistance.

Question 2: Is this hydrogel an interpenetrating network type?

Author’s Reply: Thanks for your comment! The superabsorbent composite is not an interpenetrating network type. We can conclude the 3D network structure of the superabsorbent composite was formed as the following progress: the initiator of APS was decomposed by heating and generated sulfur oxide anion radicals, and then the active groups including the hydroxyl and aldehyde group of in FA and OSSC were converted into the new radicals by electron transfer reaction. These free radicals acted as active centers and triggered the chain growth reaction in the presence of monomers AA, AM and crosslinker MBA. As the polymeric network formed completely, the FA and OSSC particles all incorporated into the polymeric network and regarded as the additional cross-linking point. FA and OSSC were introduced into the P(AA-co-AM) network structure, which can not only physically fill in the polymeric network through hydrogen bonding, but also can be chemically bonded with the polymer chain through the active groups and participated in the construction of 3D network.

Question 3: For thermal stability study author should mention the remaining char fraction amount in the text?

Author’s Reply: Thanks for your comment! According to the kind suggestion, the remaining char fraction amount has been explained at the appropriate section in the text.

Question 4: Did the author check the material’s delayed water release or deswelling in different temperature?

Author’s Reply: Thanks for your comment! The delayed water release or deswelling of superabsorbent composite in different temperature is not examined. The purpose of this work is to study the effect of the introduction non-ionic polymerization monomer AM for the salt resistance and slow release performance of superabsorbent composites. Based on previous research experience (Journal of Polymers and the Environment, 2021, 29, 4017-4026; International Journal of Biological Macromolecules, 2019, 132, 575-584; Carbohydrate Polymers, 2013, 97, 429-435), the dehydration rate of the superabsorbent composite will increase with the increase of temperature.

Question 5: What was the porosity of the hydrogels?

Author’s Reply: Thanks for your comment! Due to the superadsorbent is elastic solid and will swelling in water and the pore structure has changed obviously compared with the dry state. So discussion of the porosity is unmeaning.

Question 6: How humic acid is differed than fulvic acid regarding this delayed water release? Could the author comment on that? No need to do further experiment?

Author’s Reply: Thanks for your comment! Fulvic acid is a kind of water-soluble polymers with the smallest molecular weight, the highest physiological activity and a variety of functional groups in humic acid. Both fulvic acid and humic acid can participate in the construction of polymer 3D network through their active groups. Besides, a large number of active groups form hydrogen bonds with water molecules, so as to achieve the purpose of delaying water. However, humic acid contains other refractory substances, which can participate in the polymer network in the form of physical filling and affect the water release.

Question 7: The paper is based on superadsorbent, but there is no as such literature study on hydrogel materials. Therefore, it is recommended to insert some of relevant works.

Author’s Reply: Thanks for your comment! According to the kind suggestion, relevant literature study on hydrogel materials has been adjusted.

Reviewer 2 Report

Dear Authors,

I have a few comments relates to some section of the paaper organization, as in below, please:

L 12-15: It is a long sentence and a vague one. Please rephrase it to indicate the main problem in thiis study.

L 23-25: The main conclusion of the study should be written more clearly than this, please.

L 66-72: In this part you should mention the study objective, but not indicate methods you have used or results you have obtained.

L 92-93: Are there any references you have used for your study?

L287-292: Please split this into 2-3 sentences.

Author Response

Question 1: L12-15: It is a long sentence and a vague one. Please rephrase it to indicate the main problem in thiis study.

Author’s Reply: Thanks for your comment! The relevant content was rephrased and revised, and the corresponding position in the text is marked.

Question 2: L23-25: The main conclusion of the study should be written more clearly than this, please.

Author’s Reply: Thanks for your comment! The conclusions of relevant parts of the study were reorganized and written, and the corresponding position in the text is marked.

Question 3: L66-72: In this part you should mention the study objective, but not indicate methods you have used or results you have obtained.

Author’s Reply: Thanks for your comment! According to the kind suggestion, this part of the research method and obtained results have been deleted. The corresponding position in the text is marked.

Question 4: L92-93: Are there any references you have used for your study?

Author’s Reply: Thanks for your comment! The preparation method refers to our previous research work (Journal of Polymers and the Environment, 2021, 29, 4017-4026; International Journal of Biological Macromolecules, 2021, 179, 230-238; European Polymer Journal, 2021, 159, 110750).

Question 5: L287-292: Please split this into 2-3 sentences.

Author’s Reply: Thanks for your comment! The corresponding sentences were split and adjusted, and the corresponding position in the text is marked.

Reviewer 3 Report

  • Equation 1 must be presented as a percentage.
  • There are many typo errors in the manuscript. Need to check the writing of the manuscript.
  • In figure 5, it is suggested to present the reswelling graphs as a curve.
  • Compare all the results with other recent studies.
  • Add a section for “Future Prospects” at the end of the manuscript.
  • Some references are too old such as refs 28 and 29 and it is recommended to be replaced with newly published papers.
  • The authors can use the following references in this manuscript:

Sabbagh, F., & Muhamad, I. I. (2017). Physical and chemical characterisation of acrylamide-based hydrogels, Aam, Aam/NaCMC and Aam/NaCMC/MgO. Journal of Inorganic and Organometallic Polymers and Materials27(5), 1439-1449.

Sabbagh, F., Khatir, N. M., Karim, A. K., Omidvar, A., Nazari, Z., & Jaberi, R. (2019). Mechanical properties and swelling behavior of acrylamide hydrogels using montmorillonite and kaolinite as clays. J. Environ. Treat. Tech7, 211-219.

Author Response

Question 1: Equation 1 must be presented as a percentage.

Author’s Reply: Thanks for your comment! The expression of Equation 1 has been extensively reported in the literature (International Journal of Biological Macromolecules, 2021, 182, 1893-1905; Carbohydrate Polymers, 2021, 274, 118636; Carbohydrate Polymers, 2019, 225, 115214), which indicates the maximum amount of water that can be absorbed and retained per unit sample.

Question 2: There are many typo errors in the manuscript. Need to check the writing of the manuscript.

Author’s Reply: Thanks for your comment! The manuscript has been completely rechecked and the corresponding notes have been made in the text.

Question 3: In figure 5, it is suggested to present the reswelling graphs as a curve.

Author’s Reply: Thanks for your comment! Figure 5 has been revised according to the suggestion.

Question 4: Compare all the results with other recent studies.

Author’s Reply: Thanks for your comment! According to the kind suggestion, this study was compared with the references on slow-release fertilizers including NPK and HA in recent years (Table R1). The results indicate that poly(acrylic acid-co-acrylamide)/fulvic acid/oil shale semicoke superabsorbent composites (PAMFS) had better water absorption and salt tolerance, as well as more outstanding slow release performance. According to your kind comments, we have revised the carefully and some supporting also have been added in Table S1.

Question 5: Add a section for “Future Prospects” at the end of the manuscript.

Author’s Reply: Thanks for your comment! “Future Prospects” section was added at the end of the manuscript.

Question 6: Some references are too old such as refs 28 and 29 and it is recommended to be replaced with newly published papers.

Author’s Reply: Thanks for your comment! According to the kind suggestion, relevant literature study on hydrogel materials has been adjusted.

Round 2

Reviewer 1 Report

It can be accepted in its present form